# Intercalation-driven ferroelectric-to-ferroelastic conversion in a layered hybrid perovskite crystal

Zhenyue Wu[1,4], Shunning Li [2,4], Yasmin Mohamed Yousry [3], Walter P. D. Wong[1], Xinyun Wang[1], Teng Ma[1], Zhefeng Chen[2], Yan Shao[1], Weng Heng Liew[3], Kui Yao [3], Feng Pan [2✉] & Kian Ping Loh [1✉]

Two-dimensional (2D) organic-inorganic hybrid perovskites have attracted intense interests due to their quantum well structure and tunable excitonic properties. As an alternative to the well-studied divalent metal hybrid perovskite based on $Pb^{2+}$, $Sn^{2+}$ and $Cu^{2+}$, the trivalent metal-based (eg. $Sb^{3+}$ with ns2 outer-shell electronic configuration) hybrid perovskite with the $A_3M_2X_9$ formula (A = monovalent cations, M = trivalent metal, X = halide) offer intriguing possibilities for engineering ferroic properties. Here, we synthesized 2D ferroelectric hybrid perovskite $(TMA)_3Sb_2Cl_9$ with measurable in-plane and out-of-plane polarization. Interestingly, $(TMA)_3Sb_2Cl_9$ can be intercalated with $FeCl_4$ ions to form a ferroelastic and piezoelectric single crystal, $(TMA)_4$-Fe(iii)$Cl_4$-$Sb_2Cl_9$. Density functional theory calculations were carried out to investigate the unusual mechanism of ferroelectric-ferroelastic crossover in these crystals.

[1] Department of Chemistry, National University of Singapore, 3 Science Drive 3, Singapore 117543, Singapore. [2] School of Advanced Materials, Peking University Shenzhen Graduate School, 518055 Shenzhen, P.R. China. [3] Institute of Materials Research and Engineering, A*STAR (Agency for Science, Technology and Research), 2 Fusionopolis Way, Innovis, Singapore 138634, Singapore. [4] These authors contributed equally: Zhenyue Wu, Shunning Li. ✉email: panfeng@pkusz.edu.cn; chmlohkp@nus.edu.sg

Ferroic materials are defined by ordered states or domains that can be controlled and switched by an external field[1]. There has been an active search for materials that show ferroic properties, such as ferroelectricity and ferroelasticity because of broad applications in memory switching, shape memory, and superelastic actuators[2–8]. Two-dimensional (2D) hybrid organic-inorganic perovskites have recently emerged as a new class of ferroelectric materials, with the benefits of easy processing, structural diversity, mechanical flexibility, and intrinsic quantum-well effects[9–12]. The flexibility to tune both organic and inorganic parts in perovskite crystals enables the polar axis (direction of spontaneous polarization) to be controlled by structure and composition, such that in-plane, out-of-plane, or multi-axial (more than one axis of polarization) polarizations can be produced[13–16], thus presenting possibilities for engineering complex ferroelectric domain structures such as vortices[17].

In terms of 2D halide perovskites, one of the most effective strategies to introduce ferroelectricity is based on the so-called "geometric distortion strategy", that is the ferroelectricity is derived from rotational structural distortion of the crystalline network, and this is coupled to the order-disorder transition of the organic cations[18–20]. To achieve room-temperature ferroelectricity in 2D organic-inorganic halide perovskites, several molecular design principles have been demonstrated, such as quasi-spherical theory, homochirality, and H/F substitution[21–25]. However, the connection between ferroelectric and ferroelastic properties, in terms of engineering ferroic orders due to charge and/or strain, is not clear. To date, strategies to tune ferroic properties are developed around hybrid perovskites with divalent cations ($Pb^{2+}$, $Sn^{2+}$, $Cu^{2+}$, and $Mn^{2+}$)[26]. Trivalent cations of $Sb^{3+}$ and $Bi^{3+}$ have lone pair $ns^2$ in the outermost valence shell and resemble divalent metal cations in electronic configuration, thus they share similar geometric distortion of the inorganic octahedral and are potential candidates for engineering ferroic properties[27–30]. Trivalent 2D perovskite system generally has the formula of $A_3M_2X_9$ (A = monovalent cations, M = trivalent metal, X = halide)[31–33], and belongs to α-$Cs_3Sb_2Cl_9$-type structure[34]. The 2D analog of these perovskites can be imagined as a slab that is sliced off the (111) face of the cubic cell[35,36]. On account of the weak intermolecular interactions between neighboring layers, insertion of small molecules into the interlayer space of perovskite host lattice to impart multifunctional properties is possible[37–39]. To date, there are no reports of how ferroic properties in 2D layered hybrid perovskites can be tuned by intercalation chemistry.

Here, we demonstrate the synthesis of a ferroelectric hybrid perovskite $(TMA)_3Sb_2Cl_9$ (abbreviated as TSC, and TMA represents trimethylammonium.) with measurable in-plane and out-of-plane spontaneous polarization. More interestingly, we discover that TSC can be intercalated with trivalent metal chloride ($FeCl_4^−$) at room temperature to produce intercalation perovskite $(TMA)_4$-Fe(iii)$Cl_4$-$Sb_2Cl_9$ (TSFC) (Fig. 1a), and this is accompanied by an unusual ferroelectric-to-ferroelastic transition while keeping the piezoelectric property. Density functional theory (DFT) calculations were carried out to understand the molecular origins of ferroelectricity and ferroelasticity in TSC and TSFC, respectively. Furthermore, the shear piezoelectric coefficients in the piezoelectric matrix were measured using the laser scanning vibrometer method. This work highlights that trivalent metal-centered 2D hybrid perovskites may offer a new platform for engineering ferroic properties via intercalation chemistry.

## Results and discussion
Colorless single crystals of TSC were grown by slow evaporation of a HCl aqueous solution containing stoichiometric amounts of $Sb_2O_3$, and TMACl (see experimental section). Structure analyses of the single-crystal X-ray diffraction (XRD) data indicate that TSC crystallizes in the monoclinic polar space group $Pc$ (Tables S1 and S2). The phase purity was confirmed by powder XRD (Fig. 1c). The 2D framework of TSC resembles a (111) slice of the previously-reported inorganic perovskite α-$Cs_3Sb_2Cl_9$[40] and hybrid perovskite $(MA)_3Bi_2Br_9$[41]. Each Sb is coordinated by three bridging and three terminal Cl atoms to form the $SbCl_6$ octahedron, and the honeycomb lattice is extended by corner-sharing $SbCl_6$ octahedra lying in the $bc$-plane (Figs. 1b and S1). The geometric distortion and corrugated arrangement of $SbCl_6$ octahedra give rise to the displacement of negatively charged centers in the $ac$-plane. For the organic moieties, there are three crystallographically non-equivalent TMA cations. One cation occupies the cavity of six-membered rings formed by corner-sharing $SbCl_6$ octahedra, while the two others lie between the inorganic perovskite layers. Meanwhile, all N atoms of organic TMA cations show offset in the $ac$-plane, which means that the positively charged centers have off-centering displacement in the $ac$-plane. The synergistic motions between negatively charged inorganic $SbCl_6$ octahedra and positively charged organic cations give rise to spontaneous polarization in the $ac$-plane. This complies with the crystallographic symmetry requirement that the spontaneous polarization vector is restricted in the glide plane $σ_h$ and parallel to $ac$-plane for the ferroelectric crystal (Fig. S1), which explains the in-plane and out-of-plane polarization detected subsequently. According to Aizu notion, the symmetry breaking $2/mFm$ of TSC belongs to the uniaxial full ferroelectric/non-ferroelastic type, indicating the absence of ferroelasticity in TSC (Fig. S2 and Table S1).

The ferroelectricity of the crystal is investigated by polarization vs electric field loop carried out at room temperature, where a polarization value of about 1.76 μC/$cm^2$ is measured (Fig. 2a). To confirm the experimental result, we performed Berry phase calculation to simulate the dynamic path of the spontaneous polarization via translation, rotation, and distortion of structural element[42,43]. The calculation reveals that spontaneous polarization changes continuously from 0 to 2 μC/$cm^2$ (Fig. S3), in close agreement with experiments. The switching of the ferroelectric domains was visualized using piezoresponse force microscopy (PFM). As shown in Fig. 2b, striped ferroelectric domains and domain walls can be clearly observed in both vertical and horizontal PFM images, and oppositely polarized domains have 180° phase contrast. Ferroelectricity is observable in both the in-plane horizontal and out-of-plane vertical direction, with the consequence that the vertical PFM signals are nicely overlapped with that of horizontal PFM signals. A 180° domain wall separates adjacent domains with 180° phase difference, which confirms the uniaxial nature of TSC. Figure 2c shows a butterfly-shaped amplitude hysteresis loop collected at a single location with two sharp switching valleys; full polarization switching is observed on both left and right branches of the amplitude and phase hysteresis loops.

Interestingly, TSC can be intercalated by trivalent metal chloride (e.g., $FeCl_4^−$) to form a single-crystal TSFC that is red in color compared to the colorless TSC (Fig. 1d). Single-crystal XRD of TSFC indicates that it belongs to the orthorhombic system with $P2_12_12_1$ space group at room temperature (Tables S1 and S2); its non-centrosymmetric structure is further verified by the second harmonic generation signal (Fig. S4). The structure of intercalated TSFC can be described as alternating slabs of molecular iron chloride and antimony chloride repeating along the $c$-axis (Figs. 1a and S5). The slabs sandwiching the Fe-Cl layer in TSFC adopt the antiparallel aba′b stacking sequence. The antiparallel stacking arrangement of a and a′ breaks the symmetry glide plane $σ_h$ of the parent compound TSC and allows a $2_1$ screw axis

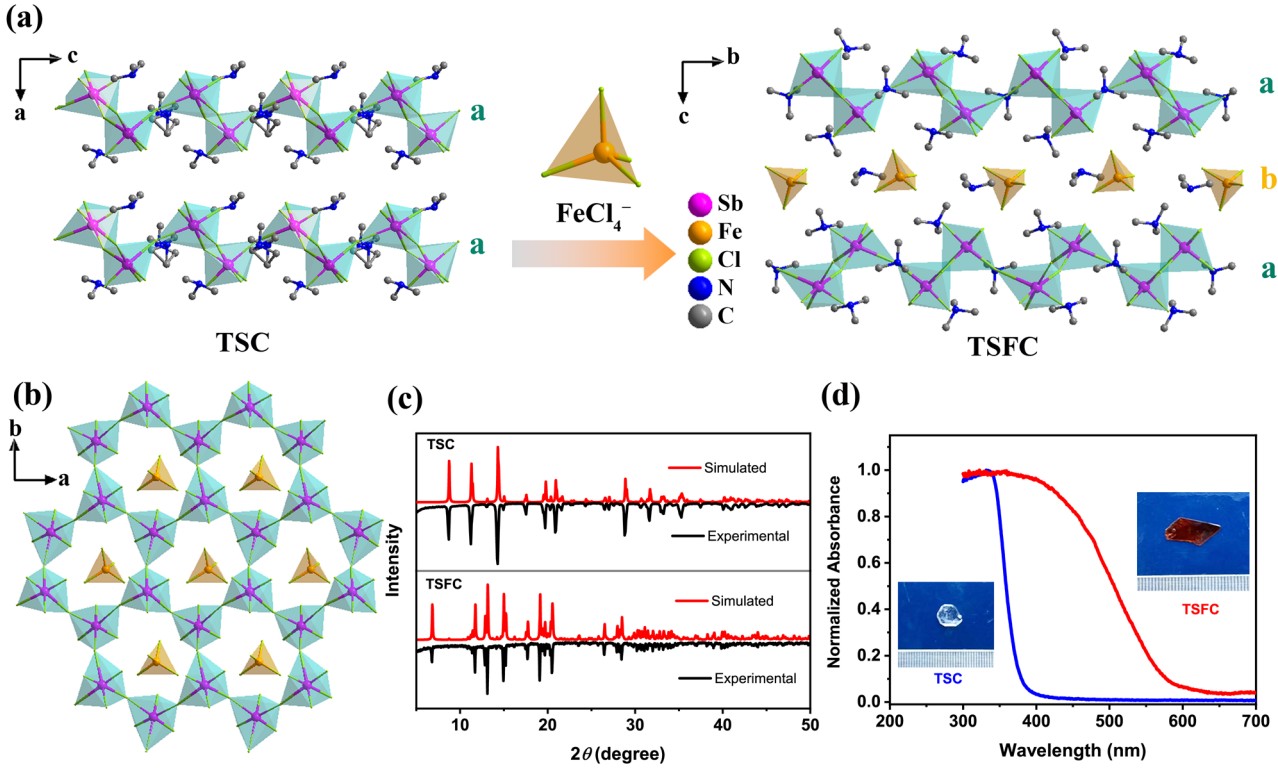

**Fig. 1 Comparison of the perovskite crystal structure of TSC and TSFC. a** Changes in stacking of the inorganic layers with FeCl₄ intercalation in TSC to form TSFC. (Turquoise a and a′ represent the honeycomb Sb-Cl layer, and orange b is intercalation Fe-Cl layer). **b** Honeycomb perovskite structure of TSFC shown along *c*-axis. All C, N, and H atoms were removed for clarity. **c** PXRD patterns for TSC and TSFC. **d** Normalized absorption spectra of TSC and TSFC. Inset: the color of growing single-crystal changes from colorless to red.

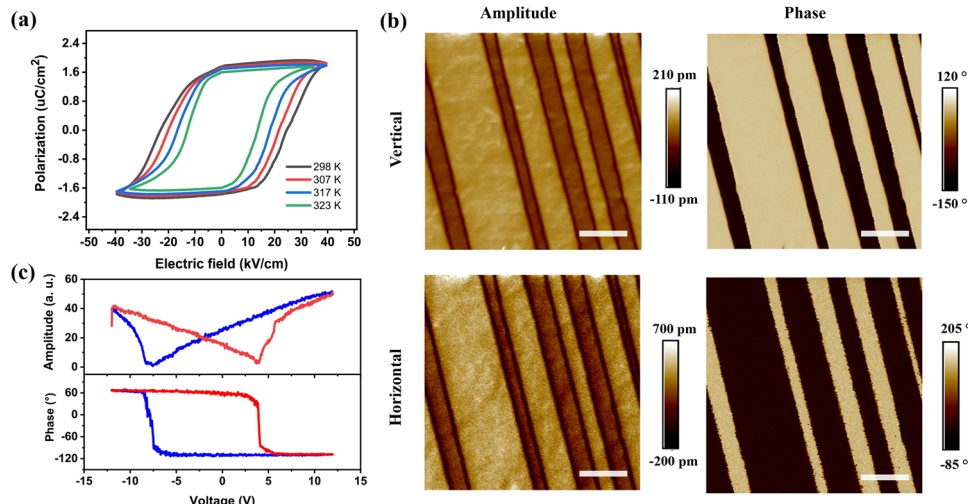

**Fig. 2 Ferroelectric hysteresis loops and domain patterns of TSC. a** Polarization vs electric field hysteresis loops at different temperature. **b** Vertical and horizontal PFM amplitude and phase signals. All scale bars are 2 μm. **c** Amplitude vs voltage and phase vs voltage curves, displaying local piezoelectric hysteresis loops.

(Figs. 1a and S6), causing the 2D intercalation perovskite to become non-ferroelectric. Due to the additional intermolecular interactions introduced by the intercalant, the unit cell of TSFC expands by 4.2% along the *b*-axis direction relative to the lattice constants of TSC, while, along *a*-axis direction, a 1.1% compression occurs. The in-plane unit cell of TSFC has expanded by 3.1% compared with TSC.

We apply polarized light optical microscopy to investigate if ferroelastic domains exist on TSFC that can exhibit spontaneous birefringence. Ferroelastic domains are known to exhibit intensity variations that accord with the distinct orientations of the domains along specific strain axes[44]. As depicted in Figs. 3a and S7, striped ferroelastic domains A, B, and C exhibit intensity contrast that is visible only under a polarized microscope, thus they are caused by different crystallographic orientations of the domains, and not due to roughness or morphologies. The angle between B striped domain and the crystallographic *b*-axis is 30°, and the same angle is observed between the C-striped domains

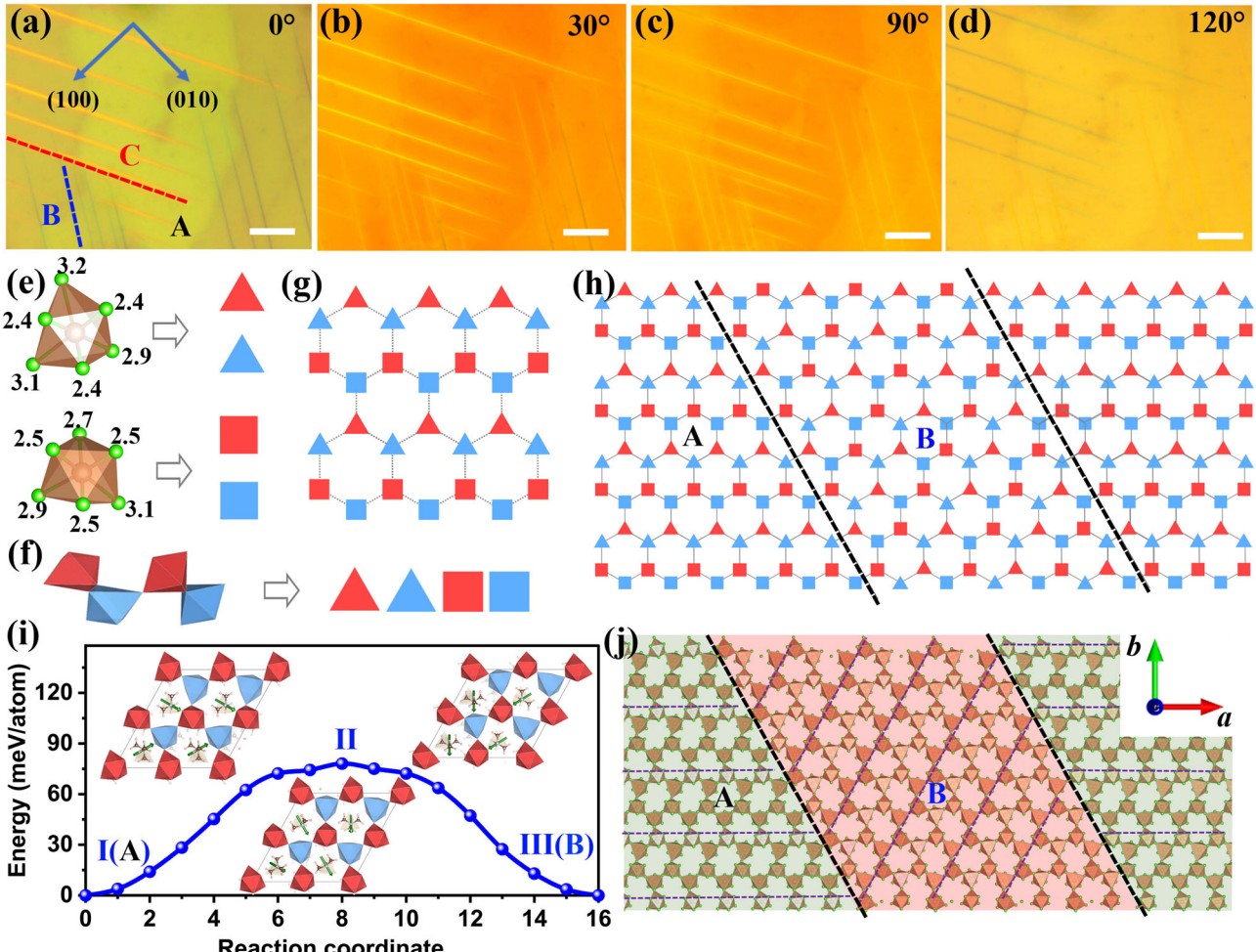

**Fig. 3 Ferroelastic domains (A, B, and C) in TSFC single crystal. a–d** Polarized microscope images for TSFC crystal obtained by rotating the polarizers at different angles. Crystallographic *a* and *b*-axis are shown. Scale bar: 20 μm. **e**, **f** Four kinds of SbCl$_6$ octahedra (red/blue triangle/square) catalogued by their distorted degree and coordinates in c direction. Topological structure of **g** the honeycomb layer and **h** domain boundaries of A and B. **i** Transformation barrier and **j** Illustrate the DFT-relaxed atomistic structures of domain boundaries formed between the A and B. Inset in **j**: the motion of molecular during the ferroelastic transition from A to B.

and the *b*-axis. The bright and dark contrasts of the domains are also visible under SEM (Fig. S8). Angle-resolved polarized optical microscopy images are shown in Figs. 3b–d and S9; the intensity contrast is caused by the changing orientation of the optical axes of the ferroelastic domain relative to the polarized light direction, thus allowing the spontaneous strain direction in each ferroelastic domain to be determined[45,46]. Based on this, we can deduce that the strain axis has $C_{3V}$ symmetry, with three equivalent strain axes that are 120° apart.

To comprehend the atomic structures of ferroelastic domains in intercalation perovskite TSFC, DFT was applied to simulate the possible types of twin boundaries. We relaxed the unit cell structure of TSFC, which is composed of two Sb-Cl layers with honeycomb architecture, two intercalated Fe-Cl layers, and a number of regularly aligned TMA molecules. Each Sb-Cl layer is composed of SbCl$_6$ corner-sharing octahedra, and the Sb atoms therein can be divided into two groups (here denoted as red and blue octahedra) according to their coordinates in *c* direction (Fig. 3f). There are two types of SbCl$_6$ octahedra with different structures, one featuring a long Sb-Cl bond (denoted as a triangle) and the other possessing a more symmetric structure (square) (Fig. 3e). Therefore, four kinds of SbCl$_6$ octahedra (red/blue triangle/square) constitute the framework of the honeycomb layer, and each one of these is aligned in the *ab*-plane (Fig. 3g). We note

that due to the honeycomb architecture, this alignment can be rotated clockwise or counterclockwise at 120° (Fig. 3h). The rotation is accomplished not by macroscopic atomic movement but by slight distortion of each SbCl$_6$ octahedron in the Sb-Cl layer, which underpins the ferroelasticity of TSFC. Different alignments correspond to different ferroelastic domains, and they can be interconnected by coherent twin boundaries dark dashed lines.

During the ferroelastic transition, FeCl$_4$ and TMA molecules undergo rotation, translation, and distortion similar to TMA in the ferroelectric transition of TSC (Fig. S10). The DFT-calculated free energy change during ferroelestic transition of TSFC via 120° clockwise rotation is shown Fig. 3i. The A variant is in the ground state, while B is strained to adopt the same supercell vectors as the A variant. The transition will encounter a barrier of 78 meV per atom, which is sufficiently low to allow facile transition between domain A and B upon the uniaxial external stress in the experiments. We note that the free energy profile for 120° counterclockwise rotation (domain C) is inverse to that of clockwise rotation, and therefore the energy barrier (A and B) is the same as the latter (A and C). As shown in Fig. 3j, the twin boundary (dark dashed line) between neighboring ferroelastic domains with two orientations (A and B) is exactly 60° relative to both directions. This feature is confirmed by experimental

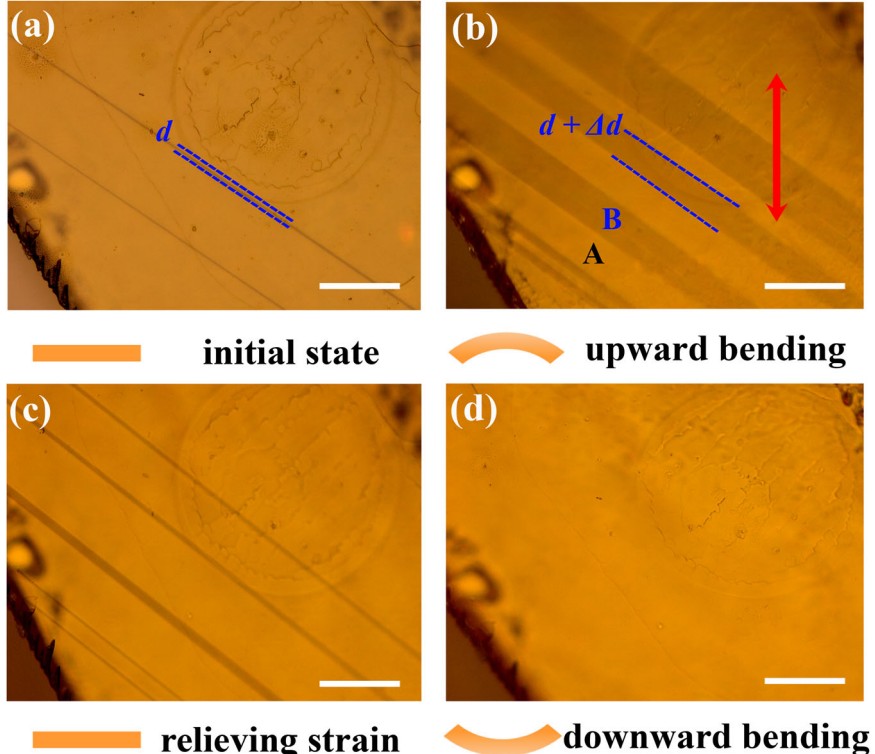

**Fig. 4 Evolution of the ferroelastic domain structure under stress-and-relax cycles. a** Polarization light image in original state of TSFC. **b–d** change in domain structure A and B under upward bending, unbend, downward bending sequence, showing hysteretic change in width of domains. The red two-way arrow represents the stress direction. *d* is the width of domain between the two blue dotted lines. Scale bar: 50 μm.

observations and can be ascribed to the honeycomb architecture of the Sb-Cl layer. The SbCl₆ octahedra at the boundary would likely take on a more symmetric shape than their original structure in TSFC, as inferred from the configuration at the energy barrier in the ferroelastic transition.

The strain-induced ferroelastic response of TSFC was investigated by subjecting the sample to bend-relax-bend cycles and observing the change in the size of the striped domains. The TSFC crystal was directly grown on a flexible substrate that allows bending. As shown in Fig. 4a, the crystal shows two stable domain variants A (bright) and B (dark). When the sample is bent upwards, the dark B domain grows at the expense of the bright A domain through the propagation of existing domain walls (Fig. 4b). Meanwhile, newly nucleated domains B are created. The expansion of B domain can be explained by the fact that its spontaneous strain direction may be aligned at a smaller angle to the direction of the applied stress[46]. After removing the external bending stress, a reversible switching from bright A domain to dark B domain occurs (Fig. 4c). However, we observed that the two dark B domains have become wider than before, such that they do not overlap with the domain patterns of the initial bending state in Fig. 4b. The hysteretic behavior is typical of ferroelastic crystals. Conversely, when the crystal is bent downwards, it is more conducive for the growth of bright A domains, and a single domain state of A can be finally formed after increasing the bending stress (Figs. 4d and S11). All these sequences confirm the dynamic switching behavior of ferroelastic domain.

To image the strain-induced switching of ferroelastic domains locally, we performed PFM imaging during, and after the application of macroscopic stress to TSFC crystal by bending the flexible substrate. Striped ferroelastic domains were observed by PFM in Figs. 5d, g and S12. That the TSFC crystal is non-ferroelectric is proven by the fact that the induced polarization is linearly proportional to the applied external electric field (Fig. S13a). Applying stress to the crystal (Fig. 5b, e, h) induces the domain to expand perpendicular to the domain walls, accompanied by two newly nucleated B domains (The change in width of the ferroelastic domains can be clearly observed by the line profile in Fig. 5b, e). After relieving the strain, the domain motion shows hysteresis, consistent with the polarized micro-scopy experiments described above (Fig. 5e, f, h, i). The height images obtained before and after stress appear unaltered (Fig. 5a–c). Therefore, both polarized light microscopy and PFM results prove that 2D intercalation perovskite TSFC is a ferroelastic material.

Next, we measured the local piezoresponse of TSFC using PFM technique. TSFC belongs to the point group of 222. According to Neumann's principle, there are 3 non-zero piezoelectric coefficients $d_{14}$, $d_{25}$, and $d_{36}$, which can be detected by recording the torsional motion of the tip cantilever in horizontal PFM mode. The tip cantilever is aligned perpendicular to the (110) direction of TSFC crystal, thus the measured piezoelectric response comes from shear piezoelectric coefficient $d_{36}$. The strongest resonance peak, which is fitted very well by the harmonic oscillator model, is obtained in TSFC by applying a 2 V tip bias (Fig. S14a). Here, to verify that the response originates from piezoelectricity of sample, resonance measurements were carried out as a function of tip bias. As depicted in Fig. S14b, the piezoresponse amplitude *vs* driving AC bias for TSFC crystals observes a linear relationship, which confirms its piezoelectricity. To measure the macroscopic piezoelectric attributes of TSFC, we use a laser scanning vibrometer to measure the displacement of the crystal along *a*, *b*, and *c*-axis of the crystal (Fig. S13b–d) after applying a unipolar

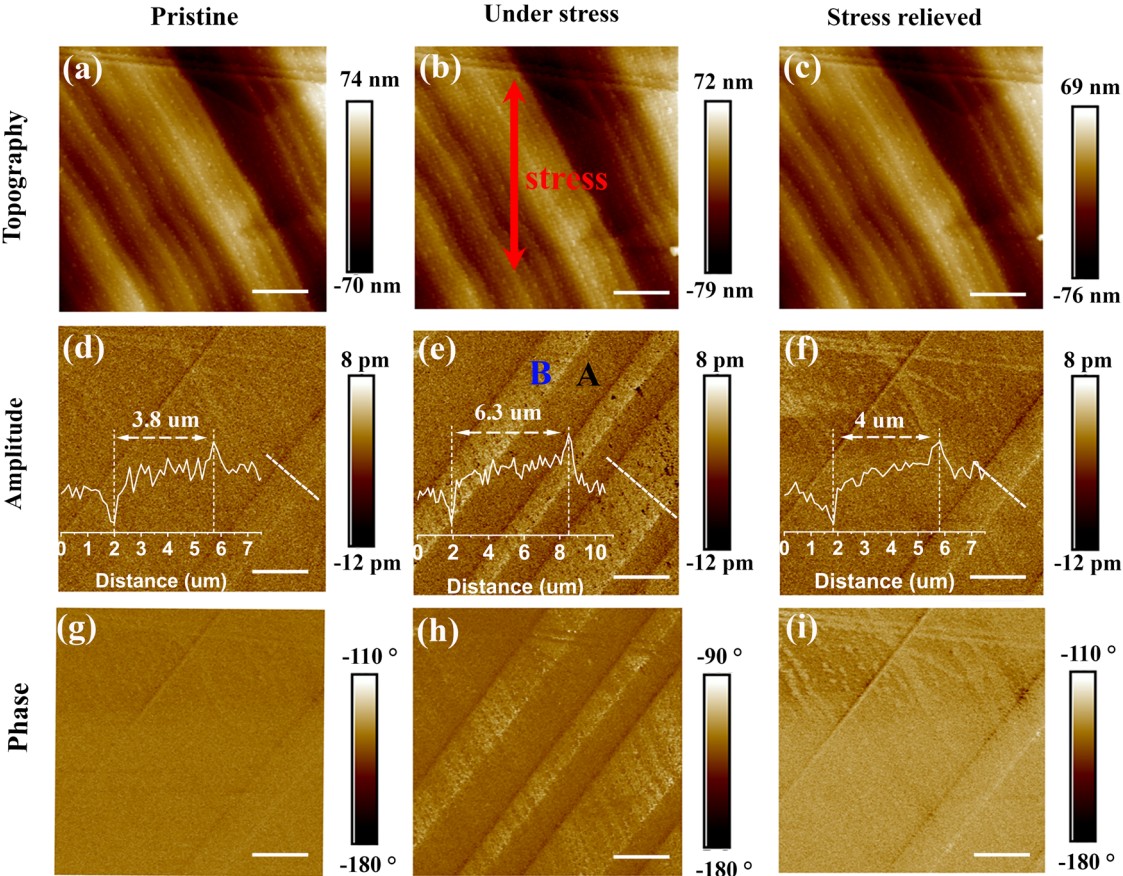

**Fig. 5 External stress modulation of ferroelastic domain patterns (A and B) observed by PFM.** Surface topography, PFM amplitude and phase images of TSFC at **a**, **d**, **g** pristine state, **b**, **e**, **h** under applying external stress, and **c**, **f**, **i** after removing the stress. The overlay plots in **d**, **e**, **f** are edge-to-edge topographical analyses of the dashed lines. The red arrow in **b** represents the direction of the applied external stress. Scale bar: 6 μm.

driving voltage at 350 kHz and amplitude of 100 V.

$$\begin{bmatrix} 0 & 0 & 0 & 7.5 & 0 & 0 \\ 0 & 0 & 0 & 0 & 6.8 & 0 \\ 0 & 0 & 0 & 0 & 0 & 9 \end{bmatrix}$$

A shear piezoelectric coefficient $d_{36}$ value of 9 pC/N is obtained along the $c$-axis, while along the $a$- and $b$-axis, the shear piezoelectric coefficients $d_{14}$ and $d_{25}$ are determined to be 7.5 and 6.8 pC/N, respectively. These piezoelectric coefficients are in the same range as that reported for piezoelectric such as $Li_2SO_4H_2O$ ($d_{25}$ ~ 5.9 pC/N)[47], $Li_2GeO_3$ ($d_{15}$ ~ 8.5 pC/N)[48], $Sr_3Ga_2Ge_4O_{14}$ ($d_{16}$ ~ 6.8 pC/N)[49], and $Pb_5Ge_3O_{11}$ ($d_{15}$ ~ 3 pC/N)[50].

In summary, we have synthesized $Sb^{3+}$-based hybrid ferroelectric TSC crystals, and reported the intercalation-induced conversion of a ferroelectric crystal (TSC) to a ferroelastic one (TSFC). Compared to the AA-stacked inorganic slabs in TSC, the antiparallel-stacked inorganic slabs in TSFC break the glide plane $\sigma_h$ prototype symmetry, leading to the loss of ferroelectricity. The in-plane 3.1% expansion of unit cells in TSFC relative to TSC creates strain axes of $C_{3V}$ symmetry and enables ferroelasticity. TSFC is also both ferroelastic and piezoelectric; its shear piezoelectric coefficients measured using the laser scanning vibrometer method are found to be comparable with that of the inorganic piezoelectric crystals. This work demonstrates a new avenue to rationally tune the ferroic properties of layered perovskite by intercalation, and paves the way forward for making multiferroic materials.

## Methods

**Synthesis and crystal growth**. Compound TSFC was prepared by mixing of trimethylamine hydrochloride (4 mmol, 0.38 g), antimony(iii) oxide (1 mmol, 0.29 g), excess iron(iii) chloride (2 mmol, 0.32 g) in concentrated aqueous HCl (47% 3 mL) in a sealed sample bottle. The compound TSC was prepared by a similar method. Bulk single crystals of these two samples were obtained by temperature-cooling technique from its saturated solution at a speed of 1 K/day. Further, 30 uL saturated solution was dropped on flexible FET and silicon substrates. Crystals grown on a substrate by a slow evaporating solvent at room temperature for 24 h were used to measure polarized microscope and PFM.

**PXRD and single-crystal structure determination**. PXRD was performed on Bruker D2 X-ray diffractometer using $Cu\ K\alpha$ radiation. Single-crystal structural data were performed on a Bruker D8 diffractometer using $Mo\ K\alpha$ radiation ($\lambda = 0.71073$). Crystal structures of TSFC and TSC were solved by direct methods and then refined by the full-matrix least-squares refinements on $F^2$ using SHELXLTL software package. Anisotropic displacement parameters were applied to all non-hydrogen atoms. All H atoms were generated by the geometrical method.

**Optical characteristics**. The optical images of ferroelastic domain patterns were captured by Olympus BX51 microscope, and a polarizer was placed in front of CCD for polarized optical study. Solid-state diffuse reflectance UV-vis spectra were measured on UV-3600 Shimadzu UV-visible spectrometer equipped with an integrating sphere and $BaSO_4$ as a reference sample.

**PFM and piezoelectricity measurements**. PFM visualization of the ferroelectric and ferroelastic domain structures was carried out using a commercial atomic force microscope system (Bruker Dimension FastScan). Conductive Pt/Ir-coated silicon probes were used for domain imaging. The drive AC amplitude of 1 V and 4 V were, respectively, applied on the tip for ferroelectric and ferroelastic to acquire PFM images. For piezoelectricity measurements, we use a laser scanning vibrometer (LSV) technique (PSV-400, PolyTech GmbH, Germany) to measure the displacement of both the electrically excited area (under the top electrode) and the substrate concurrently to determine the dilatation due to piezoelectric response. First, the bulk single crystals were cut into regular rectangles by using a scalpel

blade along crystallographic *a*, *b*, and *c*-axis directions (Fig. S13), and then both sides of these crystals along different axes were covered with silver paste electrodes. The LSV testing was conducted with an applied unipolar sine wave voltage of 100 V and 350 kHz. An area covering regions with and without the electrode pattern was scanned. The effective piezoelectric coefficient of the sample was obtained by dividing the measured displacement amplitude by the applied unipolar voltage.

**Calculation condition**. DFT calculations were performed using the Vienna Ab initio Simulation Package (VASP) with a plane-wave basis set[51,52] and the projector-augmented wave[53] pseudopotentials. Exchange-correlation interactions were treated using the generalized gradient approximation (GGA)[54] in the Perdew–Burke–Ernzerhof (PBE) form[55]. The plane-wave cutoff energy for TSC and TSFC was set to 520 and 400 eV, respectively. The Brillouin zone was sampled by Monkhorst-Pack k-point grid[56] with a total number of at least 1000/(the number of atoms per cell) points for all directions. The convergence tolerance for residual force on each atom during structure relaxation was set to 0.02 eV/Å. Spin polarization was taken into consideration and the ferromagnetic configuration was set as the initial magnetic structure. To complement the deficiency of DFT in treating dispersion interactions, a van der Waals correction term developed by Grimme (D3) was employed[57]. In order to correctly characterize the localization of transition-metal d-electrons, the GGA + U method was employed to account for the strong correlation interaction[58,59]. The value for the Hubbard U parameter of Fe was 4.0 eV[60].

**Reporting summary**. Further information on research design is available in the Nature Research Reporting Summary linked to this article.

## Data availability
All data generated and analyzed in this study are included in the Article and its Supplementary Information, and are also available from authors upon request. Crystallographic data for this paper can be obtained free of charge from the Cambridge Crystallographic Data Centre via www.ccdc.cam.ac.uk/data_request/cif. CCDC-2158201 for TSFC and CCDC-2158202 for TSC.

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

## Acknowledgements

K.P.L. would like to acknowledge Singapore's Ministry of Education Tier 2 grant MOE-T2EP50220-0002: Atomic Scale Study of Domain Switching in Two Dimensional Ferroelectrics. F.P. would like to acknowledge Soft Science Research Project of Guangdong Province (No. 2017B030301013). K.Y. would like to acknowledge partial support by A*STAR, under RIE2020 AME Individual Research Grant (IRG) (Grant No.: A20E5c0086).

## Author contributions

K.P.L. supervised the project. Z.W. designed and performed the experiments. S.L., Z.C., and F.P. performed DFT calculations. W.P.D.W. performed single-crystal XRD structure analysis. Z.W., X.W., T.M., and Y.S. performed the PFM measurement and analysis of PFM data. Y.M.Y., W.H.L., and K.Y. performed the piezoelectricity measurement. K.P.L., Z.W., and S.L. wrote the manuscript.

## Competing interests

The authors declare no competing interests.
