## [Peer Review File · Nature Communications]

REVIEWER COMMENTS

Reviewer #1 (Remarks to the Author):

In this work, Wu et al report the synthesis of a ferroelectric hybrid perovskite TSC, and by FeCl₄-intercalation they observed a transition of ferroelectric-to-ferroelastic, which is due to the intercalation induced rotation of the localized lattice units and thereby loss of ferroelectricity. The intercalation methods and likewise the resultant effect is of certain value to tune the ferroelectricity of these 2D halide perovskite materials. However, the TSC seems to be a material that has been studied before. Here are several reports studying similar composition of the A₃Sb₂X₉: e.g., *Journal of Physics: Condensed Matter*, 2003, 15, 5765.; *The Journal of Physical Chemistry A* 2005, 109, 3097-3104.; *Acta Crystallogr. Sect. B Struct. Sci.* 1996, 52, 287–295. Taking the Bi³⁺ based 2d halide perovskite, the trivalent metal-centered 2D hybrid perovskites have already been known as candidate for ferroelectric materials. From this view, the main novelty I understand of this work is the FeCl₄- intercalation methods that modulate the lattice and the consequent leading to changes of signature properties. Nevertheless, they provide a comprehensive work on how to use intercalation to modulate the materials property at a perspective of lattice and also provide calculations to explain along with this strategy, this work is of interest.

There are a few detailed technical comments:

- Line 83 “Single crystal XRD of TSFC indicates”, I didn’t see single crystal XRD data.
- The intercalation causes the lattice strain and in-plane unit cell expansion, this is the result of structure refinement analysis and observation from the experiment. What is the atomic reason for this strain? Atomic size mismatch? Or coulomb interaction or other spin-related effect? It would be great if the authors could shed some light on this.
- It seems the bandgap shrinks upon the intercalation, will this induce any leakage effect (making it more semiconductive) to the ferroelectricity measurement?
- It is suggested to add quantitative analytical plot on e.g., Fig. 5d-f, it is hard to read from the image of the 0.2 um changes.

Reviewer #2 (Remarks to the Author):

In this work, the authors report the first intercalation-induced unusual ferroic order transition from the ferroelectric TSC to ferroelastic TSFC in the 2D layered perovskite, while keeping the piezoelectricity. The ferroelectricity of TSC was confirmed by P-E loops and PFM, and the ferroelastic domain motions in TSFC

were also observed by applying external strain. Furthermore, density functional theory calculations were performed to understand the basis of ferroelectricity and ferroelasticity in TSC and TSFC. This work is interesting and important, and conclusions are well supported by experimental results. Thus, I would like to recommend publication of this work in Nature Communications after a minor revision:

1. Accompanying the paraelectric-to-ferroelectric phase transition, the paraelectric and ferroelectric phase have group-to-subgroup relationship determined by the Curie principle. This enables one to judge uniaxial or multiaxial ferroelectric by evaluating the number of equivalent polarization directions. Thus, single-crystal structure in high temperature paraelectric phase should be provided to expound “authors claim that TSC is the uniaxial ferroelectric”.
2. Curie temperature T_c is one of important parameters for ferroelectric material. The authors should provide some experimental results (such as DSC) to uncover the T_c of TSC.
3. The experimental description on piezoelectricity measurement by laser scanning vibrometer method needs much more details.
4. The coordinate axis in Figure 3j is too small to distinguish a and b-axis.
5. Some format of references (5, 11, 22 etc) should be corrected; Some important literatures on perovskite ferroelectricity and piezoelectricity (Science, 2017, 357, 306; J. Am. Chem. Soc. 2018, 140, 8051) should be cited.

Reviewer #3 (Remarks to the Author):

The manuscript by Wu et al reported the synthesis of a ferroelectric layered perovskite derivative $(TMA)_3Sb_2Cl_9$ and a ferroelastic $(TMA)_4-FeCl_4-Sb_2Cl_9$ by intercalating $(TMA)FeCl_4$ into the layer space of $(TMA)_3Sb_2Cl_9$. The single-crystal crystal structures along with ferroelectric, ferroelastic, and piezoelectric properties were well-characterized. The authors performed symmetry analysis on the two structures to understand the absence of ferroelectricity after intercalation. They also carried out DFT calculations to understand atomistic structures of ferroelastic domains in $(TMA)_4-FeCl_4-Sb_2Cl_9$. Overall, the scientific methods and results are sound. I personally feel the intercalation chemistry very interesting, which provide a new chemical strategy to tune ferroic properties of perovskite-like structures. However, I have a hard time recommending it for publication in Nature Communications based on the two following concerns.

First, while I personally feel the intercalation chemistry interesting, I do not find the presented results in the current form of particularly appealing. The authors spent a large body of discussion on the characterization of ferroelastic properties of $(TMA)_4-FeCl_4-Sb_2Cl_9$. However, the properties of this particular compound are not all that fascinating. I am not certain that these properties would be of

potential to enable disruptive applications. Based on my research background and interests, I am more curious to know more about the intercalation chemistry and its influences on the properties. For example, can the intercalation chemistry apply to other compounds or just this compound? Can the ferroic properties can be rationally tuned by the intercalating species? etc. The information is currently missing.

Second, the authors claimed in the abstract and introduction that the perovskite-like structure with the $A_3M_2X_9$ formula is a less explored system for ferroic properties. This is not correct. I have checked the literatures and quickly found several examples. To list a few,

(1) Ferroelectricity and Ferroelasticity in Organic Inorganic Hybrid (Pyrrolidinium) $_3$ [Sb $_2$ Cl $_9$], Chem. Mater. 2018, 30, 4597–4608;

(2) (C $_3$ N $_2$ H $_5$) $_3$ Sb $_2$ I $_9$ and (C $_3$ N $_2$ H $_5$) $_3$ Bi $_2$ I $_9$: ferroelastic lead-free hybrid perovskite-like materials as potential semiconducting absorbers, Dalton Trans., 2022,51, 1850-1860;

(3) Thermal, dielectric and vibrational properties of ferroelastic [(CH $_3$) $_3$ PH] $_3$ [Sb $_2$ Cl $_9$] crystal. Molecular motion of trimethylphosphonium cations studied by proton magnetic resonance., Chemical Physics 371.1-3 (2010): 66-75;

(4) Lead-free hybrid ferroelectric material based on formamidine: [NH $_2$ CHNH $_2$] $_3$ Bi $_2$ I $_9$, J. Mater. Chem. C, 2019,7, 3003-3014

These works are not appropriately acknowledged. The authors should also compare their results with these previous works and highlight their novelty.

Minor comments

1. Do the authors have the information on the phase evolution with temperature for the two structures?
2. On page 5, line 172, the authors state that “The in-plane 3.1% expansion of unit cells in TSFC relative to TSC creates strain axes of C $_3$ V symmetry and enables ferroelasticity”. Do the authors infer the absence of ferroelasticity in (TMA) $_3$ Sb $_2$ Cl $_9$?
3. On page 2, line 41, the interlayer interaction in (TMA) $_3$ Sb $_2$ Cl $_9$ is not van der Waals type.
4. In the caption of Figure 3, “ferroelectric domain patterns” should be “ferroelastic.....”

Response to reviewers

Reviewer #1 (Remarks to the Author):

In this work, Wu et al report the synthesis of a ferroelectric hybrid perovskite TSC, and by FeCl_4^- intercalation they observed a transition of ferroelectric-to-ferroelastic, which is due to the intercalation induced rotation of the localized lattice units and thereby loss of ferroelectricity. The intercalation methods and likewise the resultant effect is of certain value to tune the ferroelectricity of these 2D halide perovskite materials. However, the TSC seems to be a material that has been studied before. Here are several reports studying similar composition of the $\text{A}_3\text{Sb}_2\text{X}_9$: e.g., Journal of Physics: Condensed Matter, 2003, 15, 5765.; The Journal of Physical Chemistry A 2005, 109, 3097-3104.; Acta Crystallogr. Sect. B Struct. Sci. 1996, 52, 287–295. Taking the Bi^{3+} based 2d halide perovskite, the trivalent metal-centered 2D hybrid perovskites have already been known as candidate for ferroelectric materials. From this view, the main novelty I understand of this work is the FeCl_4^- intercalation methods that modulate the lattice and the consequent leading to changes of signature properties. Nevertheless, they provide a comprehensive work on how to use intercalation to modulate the materials property at a perspective of lattice and also provide calculations to explain along with this strategy, this work is of interest.

> We thank the reviewer for the recognition of the novelty and great support to the manuscript. Some references reviewer mentioned have been added to enrich our manuscript.

There are a few detailed technical comments:

- Line 83 “Single crystal XRD of TSFC indicates”, I didn’t see single crystal XRD data.
- > **Response:** Single crystal CIF files of TSC (CCDC number: 2158202) and TSFC (CCDC number: 2158201) have been provided in supplementary files.
- The intercalation causes the lattice strain and in-plane unit cell expansion, this is the result of structure refinement analysis and observation from the experiment. What is the atomic reason for this strain? Atomic size mismatch? Or coulomb interaction or other spin-related effect? It would be great if the authors could shed some light on this.
- > **Response:** Thanks for your useful suggestion. When Fe-Cl tetrahedron is inserted, the structural framework of the inorganic layer remains unchanged, but the arrangement of the adjacent layer changes from the parallel arrangement of the original adjacent layers to the antiparallel arrangement with 2_1 helical symmetry. This results in a significant increase in the

spacing between layers because of Fe-Cl tetrahedron with large steric hindrance. Meanwhile, Fe-Cl tetrahedron is located in the center of the upper and lower six-membered rings composed of Sb-Cl octahedra. Since the two inorganic components of Fe-Cl tetrahedron and Sb-Cl ring are both negatively charged, their mutually repulsive Coulomb interaction leads to the expansion of six-membered rings. This results in the in-plane unit cell expansion. In other words, we believe that the size mismatch and coulomb interaction is the main causes of strain in the structure.

- It seems the bandgap shrinks upon the intercalation, will this induce any leakage effect (making it more semiconductive) to the ferroelectricity measurement?

> **Response:** We use positive-up-negative-down (PUND) method to perform the P - V measurements. As shown in the following figure, the obvious leakage effect of **TSFC** for ferroelectricity measurement can be observed. Generally, for a ferroelectric material, the P - V hysteresis loop consists of three components: ferroelectric, dielectric, and leakage components in the P process. However, during the U processes, measured curve mainly includes dielectric and leakage components. These non-ferroelectric contributions were deducted by subtracting the U data from the P ones. Finally, a standard P - V hysteresis loop can be obtained. For intercalated perovskite **TSFC**, the positive branches of P and U curves are almost exactly the same, indicating the non-ferroelectric nature of **TSFC**. We only show the remanent curves of **TSFC** in the supporting information (Figure 13 a).

Positive branches of the P - V hysteresis loops measured by a positive-up-negative-down (PUND) method.

- It is suggested to add quantitative analytical plot on e.g., Fig. 5d-f, it is hard to read from the image of the 0.2 μm changes.

> **Response:** Thanks for your useful advice. We have added line profile analysis of the domain width in Fig. 5d, 5e and 5f. The amplitude signals along lines show a tip-orientation at ferroelastic domain walls, resulting from a strain gradient leading to the local structure piezoelectrically active (Nature communications, **2020**, 11, 4898). We believe that it is now

easy to read the ferroelastic domain changes from the revised images.

Figure 5. External stress modulation of ferroelastic domain patterns observed by PFM. Surface topography, PFM amplitude and phase images of TSFC at (a, d and g) pristine state, (b, e and h) under applying external stress and (c, f and i) after removing the stress. The overlay plots in d, e and f are edge-to-edge topographical analyses of the dashed lines. The red arrow in (b) represents the direction of the applied external stress.

Reviewer #2 (Remarks to the Author):

In this work, the authors report the first intercalation-induced unusual ferroic order transition from the ferroelectric TSC to ferroelastic TSFC in the 2D layered perovskite, while keeping the piezoelectricity. The ferroelectricity of TSC was confirmed by P-E loops and PFM, and the ferroelastic domain motions in TSFC were also observed by applying external strain. Furthermore, density functional theory calculations were performed to understand the basis of ferroelectricity and ferroelasticity in TSC and TSFC. This work is interesting and important, and conclusions are well supported by experimental results. Thus, I would like to recommend publication of this work in Nature Communications after a minor revision:

□ **Response:** We thank the reviewer for his or her positive comments.

1. Accompanying the paraelectric-to-ferroelectric phase transition, the paraelectric and ferroelectric phase have group-to-subgroup relationship determined by the Curie principle. This enables one to judge uniaxial or multiaxial ferroelectric by evaluating the number of equivalent polarization directions. Thus, single-crystal structure in high temperature paraelectric phase should be provided to expound “authors claim that TSC is the uniaxial ferroelectric”.

> **Response:** Thanks for reviewer’s construction advices. The high temperature has been provided in the supporting information (Figure S2), and the corresponding description have also been added.

Please see: DSC and variable-temperature dielectric constant reveal that TSC has a structure phase transition at 363 K. In high temperature (380 K) paraelectric phase, TSC crystallizes in P2₁/c space group with point group 2/m (Figure S2a). These organic TMA cations are highly disordered and have 2-fold axis symmetry. By combining the glide plane and 2-fold axis, the crystallographic inversion center of TSC is thus formed, which is consistent with the centrosymmetric structure. The symmetry breaking of TSC conforms to the Aizu notation 2/mFm, indicating that TSC is a uniaxial ferroelectric.

Figure S2. (a) DSC curves for TSC in the heating and cooling runs. (b) Variable-temperature dielectric constant of TSC in heating mode. Perspective view of TSC in high temperature paraelectric phase along (c) *b*-axis and (d) *a*-axis.

0. Curie temperature T_c is one of important parameters for ferroelectric material. The authors should provide some experimental results (such as DSC) to uncover the T_c of TSC.

> **Response:** Thanks for reviewer’s useful advices, the DSC and variable temperature dielectric results have been provided, where the peak allows us to judge the T_c of TSC. The T_c is measure to be 363 K.

1. The experimental description on piezoelectricity measurement by laser scanning vibrometer method needs much more details.

> **Response:** According to the reviewer's helpful suggestion, we have described the laser scanning vibrometer method in detail. The corresponding contents have been added in the revision.

Please see: For piezoelectricity measurements, we use a laser scanning vibrometer (LSV) technique (PSV-400, PolyTech GmbH, Germany) to measure the displacement of both the electrically excited area (under the top electrode) and the substrate concurrently to determine the dilatation due to piezoelectric response. First, the bulk single crystals were cut into regular rectangles by using a scalpel blade along crystallographic a, b, and c-axis direction (Figure S13), and then both sides of these crystals along different axes were covered with silver paste electrodes. The LSV testing was conducted with an applied unipolar sine wave voltage of 100 V and 350 kHz. An area covering regions with and without the electrode pattern was scanned. The effective piezoelectric coefficient of the sample was obtained by dividing the measured displacement amplitude by the applied unipolar voltage.

2. The coordinate axis in Figure 3j is too small to distinguish a and b-axis.

> **Response:** In the revision, we have polished a and b-axis Figure 3j. The revised figure is now clearer than before.

3. Some format of references (5, 11, 22 etc) should be corrected; Some important literatures on perovskite ferroelectricity and piezoelectricity (Science, 2017, 357, 306; J. Am. Chem. Soc. 2018, 140, 8051) should be cited.

> **Response:** We have corrected the format of references and added above-mentioned references in reference in the revised manuscript.

Reviewer #3 (Remarks to the Author):

The manuscript by Wu et al reported the synthesis of a ferroelectric layered perovskite derivative $(\text{TMA})_3\text{Sb}_2\text{Cl}_9$ and a ferroelastic $(\text{TMA})_4\text{-FeCl}_4\text{-Sb}_2\text{Cl}_9$ by intercalating $(\text{TMA})\text{FeCl}_4$ into the layer space of $(\text{TMA})_3\text{Sb}_2\text{Cl}_9$. The single-crystal crystal structures along with ferroelectric, ferroelastic, and piezoelectric properties were well-characterized. The authors performed symmetry analysis on the two structures to understand the absence of ferroelectricity after intercalation. They also carried out DFT calculations to understand atomistic structures of ferroelastic domains in $(\text{TMA})_4\text{-FeCl}_4\text{-Sb}_2\text{Cl}_9$. Overall, the scientific methods and results are sound. I personally feel the intercalation chemistry very interesting, which provide a new chemical strategy to tune ferroic

properties of perovskite-like structures. However, I have a hard time recommending it for publication in Nature Communications based on the two following concerns.

First, while I personally feel the intercalation chemistry interesting, I do not find the presented results in the current form of particularly appealing. The authors spent a large body of discussion on the characterization of ferroelastic properties of $(\text{TMA})_4\text{-FeCl}_4\text{-Sb}_2\text{Cl}_9$. However, the properties of this particular compound are not all that fascinating. I am not certain that these properties would be of potential to enable disruptive applications. Based on my research background and interests, I am more curious to know more about the intercalation chemistry and its influences on the properties. For example, can the intercalation chemistry apply to other compounds or just this compound? Can the ferroic properties can be rationally tuned by the intercalating species? etc. The information is currently missing.

- **Response:** Thanks for review's suggestion. We agree with the reviewer that the intercalation chemistry is interesting because it can modify ferroic properties. Recently, two-dimensional hybrid organic-inorganic perovskites have emerged as a new class of optoelectronic and ferroic materials with the benefits of easy processing, structural diversity, mechanical flexibility, and intrinsic quantum-well effects. Therefore, by taking advantage of the layered structure that allows intercalation by molecules, ions or atoms, some novel properties may appear, which allows the engineering of new ferroic properties. Here, our work is focused on uncovering fundamental aspect of structure chemistry and admittedly at this stage we did not show any disruptive applications. However, intercalation chemistry may allow us to uncover multiferroic properties and more research is needed.

The reason why we dedicate quite a bit of length to *present data ferroelasticity is because it is non-trivial to prove that the material is ferroelastic, as stripe-like domains that can be seen on the surfaces of these crystals are often confused with ferroelectricity*. To provide rigorous proof, that the crystal has changed from ferroelectric to ferroelastic, we need to perform the described experiments and provide the theoretical basis for it.

In terms of application, Yi et al. reported the anomalous photovoltaic effect in centrosymmetric ferroelastic BiVO_4 , where photovoltaic voltage can be reversed by stress modulation (Adv. Mater., **2018**, 30, 1870334). This is assigned to the flexoelectric coupling via a strain-induced local polarization mechanism. Here, we study the ferroelastic domain structure and atomistic arrangement for different domains, which provide the foundation of future flexoelectric memory devices. In addition, this compound also has excellent piezoelectric performance comparable to inorganic piezoelectric materials.

In addition, we have also tested the intercalation concept on some other two-dimensional

(2D) hybrid intercalation perovskites. One example of which is 2D intercalation perovskite (pyrrolidinium)₄-FeCl₄-Sb₂Cl₉ (as shown in the following picture). Without intercalation, the ferroelasticity of the host compound will be lost above 241 K, whereas intercalation of the perovskite stabilizes the ferroelasticity at room temperature. In summary, we think our work provides a new avenue to tune the ferroic properties of layered perovskite by intercalation chemistry, and pave the way forward for constructing multiferroic materials.

(a) Single crystal structural packing along *a*-axis for intercalation perovskite (pyrrolidinium)₄-FeCl₄-Sb₂Cl₉. Optical microscope images (b) without polarized light and (c) with polarized light. Scale bar 10 μm.

Second, the authors claimed in the abstract and introduction that the perovskite-like structure with the A₃M₂X₉ formula is a less explored system for ferroic properties. This is not correct. I have checked the literatures and quickly found several examples. To list a few,

- (1) Ferroelectricity and Ferroelasticity in Organic Inorganic Hybrid (Pyrrolidinium)₃[Sb₂Cl₉], Chem. Mater. 2018, 30, 4597–4608;
- (2) (C₃N₂H₅)₃Sb₂I₉ and (C₃N₂H₅)₃Bi₂I₉: ferroelastic lead-free hybrid perovskite-like materials as potential semiconducting absorbers, Dalton Trans., 2022,51, 1850-1860;
- (3) Thermal, dielectric and vibrational properties of ferroelastic [(CH₃)₃PH]₃[Sb₂Cl₉] crystal. Molecular motion of trimethylphosphonium cations studied by proton magnetic resonance., Chemical Physics 371.1-3 (2010): 66-75;
- (4) Lead-free hybrid ferroelectric material based on formamidine: [NH₂CHNH₂]₃Bi₂I₉, J. Mater. Chem. C, 2019,7, 3003-3014

These works are not appropriately acknowledged. The authors should also compare their results with these previous works and highlight their novelty.

> **Response:** Thanks for these useful suggestions, the literatures mentioned by the reviewer have been cited in the revised manuscript.

Perovskite-like structure in trivalent metal-based (Sb^{3+} , Bi^{3+}) with $\text{A}_3\text{M}_2\text{X}_9$ formula can be divided into four types: one-dimensional (1D) zig-zag double chains; two-dimensional (2D) layers; discrete bioctahedra (0D); four octahedral units (0D) ($[\text{M}_4\text{X}_{18}]^{6-}$). Within the chemical stoichiometry $\text{A}_3\text{M}_2\text{X}_9$ the ferroic properties are mainly manifested by two types of anionic sublayers: 2D and 0D-discrete bioctahedral units. The crystal structure in the literature 2, 3 and 4 adopts 0D-discrete bioctahedra framework and is not 2D. Only the compound in 1 belongs to 2D layered perovskite structure, but the fly in the ointment is that its P - E hysteresis loop could only be obtained at low temperature (less than 241 K). The standard P - E hysteresis loop of the compound **TSC** reported in this work can be obtained from 293 K to 323 K (Curie temperature \sim 363 K). Compared with well-studied divalent metal hybrid perovskite based on Pb^{2+} , Sn^{2+} and Cu^{2+} , trivalent metal-based (Sb^{3+} , Bi^{3+}) 2D perovskite systems with $\text{A}_3\text{M}_2\text{X}_9$ formula were typically stabilized with simple and small organic cations (so far, pyrrolidinium is the largest one), which limits the scope for attaining ferroic properties. Therefore, it is timely to explore ways to tune ferroic properties in trivalent metal-based 2D perovskite system. Here, by using the intercalation chemistry method, we have successfully constructed 2D intercalation perovskite, and demonstrated that the ferroic order transforms from ferroelectricity to ferroelasticity. This work not only enhances the trivalent metal-based 2D perovskite system, but also provides a new way to construct ferroic materials and further to get the multiferroic materials.

To make our motivations clearer, the description of 2D $\text{A}_3\text{M}_2\text{X}_9$ perovskites in the abstract and introduction have been revised (see the highlighted sections).

Minor comments

1. Do the authors have the information on the phase evolution with temperature for the two structures?

> **Response:** We have performed DSC to study the phase change of these two compounds. DSC curves reveal **TSC** and **TSFC** have structure phase transition at 363 K and 325 K, respectively. **TSC** is the ferroelectric phase at room temperature and crystallizes in Pc space group and point group of m ; while at the high temperature paraelectric phase, the space group changes from Pc to $P2_1/c$. According to Aizu notion, the symmetry breaking of **TSC** is $2/mFm$, pertaining to the 88 potential ferroelectric phase transitions. For ferroelastic **TSFC**, it belongs to $P2_12_12_1$ and $C222_1$ space group at 293 K and 335 K respectively. In the high temperature 335K, we still can observe the stripe ferroelastic domain under polarized light

microscope, indicating **TSFC** occurs ferroelastic-to-ferroelastic phase transforms. We did not observe the paraelastic phase until the sample is heated to 395 K (melting point).

DSC curves of (a) **TSC** and (d) **TSFC** in the heating and cooling runs. Single crystal structure of **TSC** in the (b) room temperature ferroelectric phase and (c) high temperature paraelectric phase. Ferroelastic phase structure of **TSFC** at the (e) room temperature and (f) high temperature.

2. On page 5, line 172, the authors state that “The in-plane 3.1% expansion of unit cells in TSFC relative to TSC creates strain axes of C_{3V} symmetry and enables ferroelasticity”. Do the authors infer the absence of ferroelasticity in $(TMA)_3Sb_2Cl_9$?

> **Response:** $(TMA)_3Sb_2Cl_9$ (**TSC**) in the high temperature paraelectric phase and room temperature ferroelectric phase belong to $2/m$ ($P2_1/c$) and m (Pc) point group, respectively. According to Aizu notion, the symmetry breaking $2/mFm$ of **TSC** belongs to full ferroelectric/non-ferroelastic type (Phys. Rev. B, **1970**, 2, 754), indicating the absence of ferroelasticity in $(TMA)_3Sb_2Cl_9$.

3. On page 2, line 41, the interlayer interaction in $(TMA)_3Sb_2Cl_9$ is not van der Waals type.

> **Response:** line 41 have been revised.

Please see: On account of the weak intermolecular interactions between neighboring layers, insertion of small molecules into the interspace of perovskite host lattice to impart multifunctional properties may be possible.

4. In the caption of Figure 3, “ferroelectric domain patterns” should be “ferroelastic.....”

> **Response:** We have changed the “ferroelectric” to “ferroelastic” in the caption of Figure 3.

REVIEWERS' COMMENTS

Reviewer #1 (Remarks to the Author):

The authors have answered all my questions. I think the current version is ready to publish.

Reviewer #2 (Remarks to the Author):

The raised points have been well solved. According to this reviewer's judgment, the present work will be an outstanding and exciting contribution to ferroelectricity and related fields. Thus, this reviewer does recommend the publication of the present article in Nature Communications without further changes.

Reviewer #3 (Remarks to the Author):

The authors have addressed my comments. I recommend publication of the work

Response to reviewers

Reviewer #1 (Remarks to the Author):

The authors have answered all my questions. I think the current version is ready to publish.

- **Response:** We thank the reviewer for this recommendation.

Reviewer #2 (Remarks to the Author):

The raised points have been well solved. According to this reviewer's judgment, the present work will be an outstanding and exciting contribution to ferroelectricity and related fields. Thus, this reviewer does recommend the publication of the present article in Nature Communications without further changes.

- **Response:** We thank the reviewer for this recommendation.

Reviewer #3 (Remarks to the Author):

The authors have addressed my comments. I recommend publication of the work.

- **Response:** We thank the reviewer for this recommendation.